# A Simple Practical Accelerated Method for Finite Sums

**Aaron Defazio**
Ambiata, Sydney Australia

## Abstract

We describe a novel optimization method for finite sums (such as empirical risk minimization problems) building on the recently introduced SAGA method. Our method achieves an accelerated convergence rate on strongly convex smooth problems. Our method has only one parameter (a step size), and is radically simpler than other accelerated methods for finite sums. Additionally it can be applied when the terms are non-smooth, yielding a method applicable in many areas where operator splitting methods would traditionally be applied.

## Introduction

A large body of recent developments in optimization have focused on minimization of convex finite sums of the form:

$$f(x) = \frac{1}{n} \sum_{i=1}^{n} f_i(x),$$

a very general class of problems including the empirical risk minimization (ERM) framework as a special case. Any function $h$ can be written in this form by setting $f_1(x) = h(x)$ and $f_i = 0$ for $i \neq 1$, however when each $f_i$ is sufficiently regular in a way that can be made precise, it is possible to optimize such sums more efficiently than by treating them as black box functions.

In most cases recently developed methods such as SAG [Schmidt et al., 2013] can find an $\epsilon$-minimum faster than either stochastic gradient descent or accelerated black-box approaches, both in theory and in practice. We call this class of methods fast incremental gradient methods (FIG).

FIG methods are randomized methods similar to SGD, however unlike SGD they are able to achieve linear convergence rates under Lipschitz-smooth and strong convexity conditions [Mairal, 2014, Defazio et al., 2014b, Johnson and Zhang, 2013, Konečný and Richtárik, 2013]. The linear rate in the first wave of FIG methods directly depended on the condition number $L/\mu$ of the problem, whereas recently several methods have been developed that depend on the square-root of the condition number [Lan and Zhou, 2015, Lin et al., 2015, Shalev-Shwartz and Zhang, 2013c, Nitanda, 2014], at least when $n$ is not too large. Analogous to the black-box case, these methods are known as accelerated methods.

In this work we develop another accelerated method, which is conceptually simpler and requires less tuning than existing accelerated methods. The method we give is a primal approach, however it makes use of a proximal operator oracle for each $f_i$ instead of a gradient oracle, unlike other primal approaches. The proximal operator is also used by dual methods such as some variants of SDCA [Shalev-Shwartz and Zhang, 2013a].

**Algorithm 1**

Pick some starting point $x^0$ and step size $\gamma$. Initialize each $g_i^0 = f_i'(x^0)$, where $f_i'(x^0)$ is any gradient/subgradient at $x^0$.

Then at step $k+1$:

1. Pick index $j$ from 1 to $n$ uniformly at random.

2. Update $x$:

$$z_j^k = x^k + \gamma \left[ g_j^k - \frac{1}{n} \sum_{i=1}^{n} g_i^k \right],$$

$$x^{k+1} = \text{prox}_j^\gamma \left( z_j^k \right).$$

3. Update the gradient table: Set $g_j^{k+1} = \frac{1}{\gamma} \left( z_j^k - x^{k+1} \right)$, and leave the rest of the entries unchanged ($g_i^{k+1} = g_i^k$ for $i \neq j$).

---

## 1    Algorithm

Our algorithm's main step makes use of the proximal operator for a randomly chosen $f_i$. For convenience, we define:

$$\text{prox}_i^\gamma (x) = \text{argmin}_y \left\{ \gamma f_i(y) + \frac{1}{2} \|x - y\|^2 \right\}.$$

This proximal operator can be computed efficiently or in closed form in many cases, see Section 4 for details. Like SAGA, we also maintain a table of gradients $g_i$, one for each function $f_i$. We denote the state of $g_i$ at the end of step $k$ by $g_i^k$. The iterate (our guess at the solution) at the end of step $k$ is denoted $x^k$. The starting iterate $x^0$ may be chosen arbitrarily.

The full algorithm is given as Algorithm 1. The sum of gradients $\frac{1}{n} \sum_{i=1}^{n} g_i^k$ can be cached and updated efficiently at each step, and in most cases instead of storing a full vector for each $g_i$, only a single real value needs to be stored. This is the case for linear regression or binary classification with logistic loss or hinge loss, in precisely the same way as for standard SAGA. A discussion of further implementation details is given in Section 4.

With step size

$$\gamma = \frac{\sqrt{(n-1)^2 + 4n\frac{L}{\mu}}}{2Ln} - \frac{1 - \frac{1}{n}}{2L},$$

the expected convergence rate in terms of squared distance to the solution is given by:

$$E \left\| x^k - x^* \right\|^2 \leq \left( 1 - \frac{\mu\gamma}{1 + \mu\gamma} \right)^k \frac{\mu + L}{\mu} \left\| x^0 - x^* \right\|^2,$$

when each $f_i : \mathbb{R}^d \to \mathbb{R}$ is $L$-smooth and $\mu$-strongly convex. See Nesterov [1998] for definitions of these conditions. Using big-O notation, the number of steps required to reduce the distance to the solution by a factor $\epsilon$ is:

$$k = O \left( \left( \sqrt{\frac{nL}{\mu}} + n \right) \log \left( \frac{1}{\epsilon} \right) \right),$$

as $\epsilon \to 0$. This rate matches the lower bound known for this problem [Lan and Zhou, 2015] under the gradient oracle. We conjecture that this rate is optimal under the proximal operator oracle as well. Unlike other accelerated approaches though, we have only a single tunable parameter (the step size $\gamma$), and the algorithm doesn't need knowledge of $L$ or $\mu$ except for their appearance in the step size.

Compared to the $O\left((L/\mu + n) \log (1/\epsilon)\right)$ rate for SAGA and other non-accelerated FIG methods, accelerated FIG methods are significantly faster when $n$ is small compared to $L/\mu$, however for $n \geq L/\mu$ the performance is essentially the same. All known FIG methods hit a kind of wall at $n \approx L/\mu$, where they decrease the error at each step by no more than $1 - \frac{1}{n}$. Indeed, when $n \geq L/\mu$ the problem is so well conditioned so as to be easy for any FIG method to solve it efficiently. This is sometimes called the big data setting [Defazio et al., 2014b].

Our convergence rate can also be compared to that of optimal first-order black box methods, which have rates of the form $k = O\left(\left(\sqrt{L/\mu}\right)\log\left(1/\epsilon\right)\right)$ per epoch equivalent. We are able to achieve a $\sqrt{n}$ speedup on a per-epoch basis, for $n$ not too large. Of course, all of the mentioned rates are significantly better than the $O\left(\left(L/\mu\right)\log\left(1/\epsilon\right)\right)$ rate of gradient descent.

For non-smooth but strongly convex problems, we prove a $1/\epsilon$-type rate under a standard iterate averaging scheme. This rate does not require the use of decreasing step sizes, so our algorithm requires less tuning than other primal approaches on non-smooth problems.

## 2 Relation to other approaches

Our method is most closely related to the SAGA method. To make the relation clear, we may write our method's main step as:

$$x^{k+1} = x^k - \gamma\left[f_j'(x^{k+1}) - g_j^k + \frac{1}{n}\sum_{i=1}^{n} g_i^k\right],$$

whereas SAGA has a step of the form:

$$x^{k+1} = x^k - \gamma\left[f_j'(x^k) - g_j^k + \frac{1}{n}\sum_{i=1}^{n} g_i^k\right].$$

The difference is the point at which the gradient of $f_j$ is evaluated at. The proximal operator has the effect of evaluating the gradient at $x^{k+1}$ instead of $x^k$. While a small difference on the surface, this change has profound effects. It allows the method to be applied directly to non-smooth problems using fixed step sizes, a property not shared by SAGA or other primal FIG methods. Additionally, it allows for much larger step sizes to be used, which is why the method is able to achieve an accelerated rate.

It is also illustrative to look at how the methods behave at $n = 1$. SAGA degenerates into regular gradient descent, whereas our method becomes the proximal-point method [Rockafellar, 1976]:

$$x^{k+1} = \text{prox}_{\gamma f}(x^k).$$

The proximal point method has quite remarkable properties. For strongly convex problems, it converges *for any* $\gamma > 0$ at a linear rate. The downside being the inherent difficulty of evaluating the proximal operator. For the $n = 2$ case, if each term is an indicator function for a convex set, our algorithm matches Dykstra's projection algorithm if we take $\gamma = 2$ and use cyclic instead of random steps.

#### Accelerated incremental gradient methods

Several acceleration schemes have been recently developed as extensions of non-accelerated FIG methods. The earliest approach developed was the ASDCA algorithm [Shalev-Shwartz and Zhang, 2013b,c]. The general approach of applying the proximal-point method as the outer-loop of a double-loop scheme has been dubbed the Catalyst algorithm Lin et al. [2015]. It can be applied to accelerate any FIG method. Recently a very interesting primal-dual approach has been proposed by Lan and Zhou [2015]. All of the prior accelerated methods are significantly more complex than the approach we propose, and have more complex proofs.

## 3 Theory

### 3.1 Proximal operator bounds

In this section we rehash some simple bounds from proximal operator theory that we will use in this work. Define the short-hand $p_{\gamma f}(x) = \text{prox}_{\gamma f}(x)$, and let $g_{\gamma f}(x) = \frac{1}{\gamma}\left(x - p_{\gamma f}(x)\right)$, so that $p_{\gamma f}(x) = x - \gamma g_{\gamma f}(x)$. Note that $g_{\gamma f}(x)$ is a subgradient of $f$ at the point $p_{\gamma f}(x)$. This relation is known as the **optimality condition** of the proximal operator. Note that proofs for the following two propositions are in the supplementary material.

| Notation | Description | Additional relation |
|---|---|---|
| $x^k$ | Current iterate at step $k$ | $x^k \in R^d$ |
| $x^*$ | Solution | $x^* \in R^d$ |
| $\gamma$ | Step size | |
| $p_{\gamma f}(x)$ | Short-hand in results for generic $f$ | $p_{\gamma f}(x) = \text{prox}_{\gamma f}(x)$ |
| $\text{prox}_i^\gamma(x)$ | Proximal operator of $\gamma f_i$ at $x$ | $= \text{argmin}_y \left\{ \gamma f_i(y) + \frac{1}{2} \|x - y\|^2 \right\}$ |
| $g_i^k$ | A stored subgradient of $f_i$ as seen at step $k$ | |
| $g_i^*$ | A subgradient of $f_i$ at $x^*$ | $\sum_{i=1}^n g_i^* = 0$ |
| $v_i$ | $v_i = x^* + \gamma g_i^*$ | $x^* = \text{prox}_i^\gamma(v_i)$ |
| $j$ | Chosen component index (random variable) | |
| $z_j^k$ | $z_j^k = x^k + \gamma \left[ g_j^k - \frac{1}{n} \sum_{i=1}^n g_i^k \right]$ | $x_j^{k+1} = \text{prox}_j^\gamma \left( z_j^k \right)$ |

Table 1: Notation quick reference

**Proposition 1.** *(**Strengthening firm non-expansiveness under strong convexity**) For any $x, y \in \mathbb{R}^d$, and any convex function $f : \mathbb{R}^d \to \mathbb{R}$ with strong convexity constant $\mu \geq 0$,*

$$\langle x - y, p_{\gamma f}(x) - p_{\gamma f}(y) \rangle \geq (1 + \mu\gamma) \|p_{\gamma f}(x) - p_{\gamma f}(y)\|^2 .$$

*In operator theory this property is known as $(1 + \mu\gamma)$-cocoerciveness of $p_{\gamma f}$.*

**Proposition 2.** *(**Moreau decomposition**) For any $x \in \mathbb{R}^d$, and any convex function $f : \mathbb{R}^d \to \mathbb{R}$ with Fenchel conjugate $f^*$ :*

$$p_{\gamma f}(x) = x - \gamma p_{\frac{1}{\gamma} f^*}(x/\gamma). \tag{1}$$

*Recall our definition of $g_{\gamma f}(x) = \frac{1}{\gamma}(x - p_{\gamma f}(x))$ also. After combining, the following relation thus holds between the proximal operator of the conjugate $f^*$ and $g_{\gamma f}$:*

$$p_{\frac{1}{\gamma} f^*}(x/\gamma) = \frac{1}{\gamma}(x - p_{\gamma f}(x)) = g_{\gamma f}(x). \tag{2}$$

**Theorem 3.** *For any $x, y \in \mathbb{R}^d$, and any convex L-smooth function $f : \mathbb{R}^d \to \mathbb{R}$:*

$$\langle g_{\gamma f}(x) - g_{\gamma f}(y), x - y \rangle \geq \gamma \left( 1 + \frac{1}{L\gamma} \right) \|g_{\gamma f}(x) - g_{\gamma f}(y)\|^2 ,$$

*Proof.* We will apply cocoerciveness of the proximal operator of $f^*$ as it appears in the decomposition. Note that L-smoothness of $f$ implies $1/L$-strong convexity of $f^*$. In particular we apply it to the points $\frac{1}{\gamma} x$ and $\frac{1}{\gamma} y$:

$$\left\langle p_{\frac{1}{\gamma} f^*}(\frac{1}{\gamma} x) - p_{\frac{1}{\gamma} f^*}(\frac{1}{\gamma} y), \frac{1}{\gamma} x - \frac{1}{\gamma} y \right\rangle \geq \left( 1 + \frac{1}{L\gamma} \right) \left\| p_{\frac{1}{\gamma} f^*}(\frac{1}{\gamma} x) - p_{\frac{1}{\gamma} f^*}(\frac{1}{\gamma} y) \right\|^2 .$$

Pulling $\frac{1}{\gamma}$ from the right side of the inner product out, and plugging in Equation 2, gives the result. □

### 3.2 Notation

Let $x^*$ be the unique minimizer (due to strong convexity) of $f$. In addition to the notation used in the description of the algorithm, we also fix a set of subgradients $g_j^*$, one for each of $f_j$ at $x^*$, chosen such that $\sum_{j=1}^n g_j^* = 0$. We also define $v_j = x^* + \gamma g_j^*$. Note that at the solution $x^*$, we want to apply a proximal step for component $j$ of the form:

$$x^* = \text{prox}_j^\gamma \left( x^* + \gamma g_j^* \right) = \text{prox}_j^\gamma \left( v_j \right) .$$

**Lemma 4.** *(Technical lemma needed by main proof) Under Algorithm 1, taking the expectation over the random choice of $j$, conditioning on $x^k$ and each $g_i^k$, allows us to bound the following inner product at step $k$:*

$$E \left\langle \gamma \left[ g_j^k - \frac{1}{n} \sum_{i=1}^{n} g_i^k \right] - \gamma g_j^*, (x^k - x^*) + \gamma \left[ g_j^k - \frac{1}{n} \sum_{i=1}^{n} g_i^k \right] - \gamma g_j^* \right\rangle$$

$$\leq \gamma^2 \frac{1}{n} \sum_{i=1}^{n} \left\| g_i^k - g_i^* \right\|^2.$$

*The proof is in the supplementary material.*

### 3.3   Main result

**Theorem 5.** *(single step Lyapunov descent) We define the Lyapunov function $T^k$ of our algorithm (Point-SAGA) at step $k$ as:*

$$T^k = \frac{c}{n} \sum_{i=1}^{n} \left\| g_i^k - g_i^* \right\|^2 + \left\| x^k - x^* \right\|^2,$$

*for $c = 1/\mu L$. Then using step size $\gamma = \frac{\sqrt{(n-1)^2 + 4n\frac{L}{\mu}}}{2Ln} - \frac{1 - \frac{1}{n}}{2L}$, the expectation of $T^{k+1}$, over the random choice of $j$, conditioning on $x^k$ and each $g_i^k$, is:*

$$E\left[T^{k+1}\right] \leq (1 - \kappa) T^k \quad \text{for } \kappa = \frac{\mu\gamma}{1 + \mu\gamma},$$

*when each $f_i : \mathbb{R}^d \to \mathbb{R}$ is $L$-smooth and $\mu$-strongly convex and $0 < \mu < L$. This is the same Lyapunov function as used by Hofmann et al. [2015].*

*Proof.* Term 1 of $T^{k+1}$ is straight-forward to simplify:

$$\frac{c}{n} E \sum_{i=1}^{n} \left\| g_i^{k+1} - g_i^* \right\|^2 = \left(1 - \frac{1}{n}\right) \frac{c}{n} \sum_{i=1}^{n} \left\| g_i^k - g_i^* \right\|^2 + \frac{c}{n} E \left\| g_j^{k+1} - g_j^* \right\|^2.$$

For term 2 of $T^{k+1}$ we start by applying cocoerciveness (Theorem 1):

$$(1 + \mu\gamma) E \left\| x^{k+1} - x^* \right\|^2$$

$$= (1 + \mu\gamma) E \left\| \text{prox}_j^\gamma(z_j^k) - \text{prox}_j^\gamma(v_j) \right\|^2$$

$$\leq E \left\langle \text{prox}_j^\gamma(z_j^k) - \text{prox}_j^\gamma(v_j), z_j^k - v_j \right\rangle$$

$$= E \left\langle x^{k+1} - x^*, z_j^k - v_j \right\rangle.$$

Now we add and subtract $x^k$ :

$$= E \left\langle x^{k+1} - x^k + x^k - x^*, z_j^k - v_j \right\rangle$$

$$= E \left\langle x^k - x^*, z_j^k - v_j \right\rangle + E \left\langle x^{k+1} - x^k, z_j^k - v_j \right\rangle$$

$$= \left\| x^k - x^* \right\|^2 + E \left\langle x^{k+1} - x^k, z_j^k - v_j \right\rangle,$$

where we have pulled out the quadratic term by using $E[z_j^k - v_j] = x^k - x^*$ (we can take the expectation since the left hand side of the inner product doesn't depend on $j$). We now expand $E \left\langle x^{k+1} - x^k, z_j^k - v_j \right\rangle$ further:

$$E \left\langle x^{k+1} - x^k, z_j^k - v_j \right\rangle$$

$$= E \left\langle x^{k+1} - \gamma g_j^* + \gamma g_j^* - x^k, z_j^k - v_j \right\rangle$$

$$= E \left\langle x^k - \gamma g_j^{k+1} + \gamma \left[ g_j^k - \frac{1}{n} \sum_{i=1}^{n} g_i^k \right] - \gamma g_j^* + \gamma g_j^* - x^k, \right.$$

$$\left. (x^k - x^*) + \gamma \left[ g_j^k - \frac{1}{n} \sum_{i=1}^{n} g_i^k \right] - \gamma g_j^* \right\rangle. \tag{3}$$

We further split the left side of the inner product to give two separate inner products:

$$= E \left\langle \gamma \left[ g_j^k - \frac{1}{n} \sum_{i=1}^n g_i^k \right] - \gamma g_j^*, \left( x^k - x^* \right) + \gamma \left[ g_j^k - \frac{1}{n} \sum_{i=1}^n g_i^k \right] - \gamma g_j^* \right\rangle$$

$$+ E \left\langle \gamma g_j^* - \gamma g_j^{k+1}, \left( x^k - x^* \right) + \gamma \left[ g_j^k - \frac{1}{n} \sum_{i=1}^n g_i^k \right] - \gamma g_j^* \right\rangle. \tag{4}$$

The first inner product in Equation 4 is the quantity we bounded in Lemma 4 by $\gamma^2 \frac{1}{n} \sum_{i=1}^n \left\| g_i^k - g_i^* \right\|^2$. The second inner product in Equation 4, can be simplified using Theorem 3 (note the right side of the inner product is equal to $z_j^k - v_j$):

$$-\gamma E \left\langle g_j^{k+1} - g_j^*, \, z_j^k - v_j \right\rangle \leq -\gamma^2 \left( 1 + \frac{1}{L\gamma} \right) E \left\| g_j^{k+1} - g_j^* \right\|^2.$$

Combing these gives the following bound on $(1 + \mu\gamma) E \left\| x^{k+1} - x^* \right\|^2$:

$$(1+\mu\gamma) E \left\| x^{k+1} - x^* \right\|^2 \leq \left\| x^k - x^* \right\|^2 + \gamma^2 \frac{1}{n} \sum_{i=1}^n \left\| g_i^k - g_i^* \right\|^2 - \gamma^2 \left( 1 + \frac{1}{L\gamma} \right) E \left\| g_j^{k+1} - g_j^* \right\|^2.$$

Define $\alpha = \frac{1}{1+\mu\gamma} = 1 - \kappa$, where $\kappa = \frac{\mu\gamma}{1+\mu\gamma}$. Now we multiply the above inequality through by $\alpha$ and combine with the rest of the Lyapunov function, giving:

$$E \left[ T^{k+1} \right] \leq T^k + \left( \alpha\gamma^2 - \frac{c}{n} \right) \frac{1}{n} \sum_i^n \left\| g_i^k - g_i^* \right\|^2$$

$$+ \left( \frac{c}{n} - \alpha\gamma^2 - \frac{\alpha\gamma}{L} \right) E \left\| g_j^{k+1} - g_j^* \right\|^2 - \kappa E \left\| x^k - x^* \right\|^2.$$

We want an $\alpha$ convergence rate, so we pull out the required terms:

$$E \left[ T^{k+1} \right] \leq \alpha T^k + \left( \alpha\gamma^2 + \kappa c - \frac{c}{n} \right) \frac{1}{n} \sum_i^n \left\| g_i^k - g_i^* \right\|^2$$

$$+ \left( \frac{c}{n} - \alpha\gamma^2 - \frac{\alpha\gamma}{L} \right) E \left\| g_j^{k+1} - g_j^* \right\|^2.$$

Now to complete the proof we note that $c = 1/\mu L$ and $\gamma = \frac{\sqrt{(n-1)^2 + 4n\frac{L}{\mu}}}{2Ln} - \frac{1 - \frac{1}{n}}{2L}$ ensure that both terms inside the round brackets are non-positive, giving $E T^{k+1} \leq \alpha T^k$. These constants were found by equating the equations in the brackets to zero, and solving with respect to the two unknowns, $\gamma$ and $c$. It is easy to verify that $\gamma$ is always positive, as a consequence of the condition number $L/\mu$ always being at least 1. □

**Corollary 6.** *(Smooth case) Chaining Theorem 5 gives a convergence rate for Point-SAGA at step $k$ under the constants given in Theorem 5 of:*

$$E \left\| x^k - x^* \right\|^2 \leq (1 - \kappa)^k \frac{\mu + L}{\mu} \left\| x^0 - x^* \right\|^2,$$

*if each $f_i : \mathbb{R}^d \to \mathbb{R}$ is $L$-smooth and $\mu$-strongly convex.*

**Theorem 7.** *(Non-smooth case) Suppose each $f_i : \mathbb{R}^d \to \mathbb{R}$ is $\mu$-strongly convex, $\left\| g_i^0 - g_i^* \right\| \leq B$ and $\left\| x^0 - x^* \right\| \leq R$. Then after $k$ iterations of Point-SAGA with step size $\gamma = R/B\sqrt{n}$:*

$$E \left\| \bar{x}^k - x^* \right\|^2 \leq 2 \frac{\sqrt{n} \left( 1 + \mu \left( R/B\sqrt{n} \right) \right)}{\mu k} RB,$$

*where $\bar{x}^k = \frac{1}{k} E \sum_{t=1}^k x^t$. The proof of this theorem is included in the supplementary material.*

# 4 Implementation

Care must be taken for efficient implementation, particularly in the sparse gradient case. We discuss the key points below. A fast Cython implementation is available on the author's website incorporating these techniques.

**Proximal operators** For the most common binary classification and regression methods, implementing the proximal operator is straight-forward. We include details of the computation of the proximal operators for the hinge, square and logistic losses in the supplementary material. The logistic loss does not have a closed form proximal operator, however it may be computed very efficiently in practice using Newton's method on a 1D subproblem. For problems of a non-trivial dimensionality the cost of the dot products in the main step is much greater than the cost of the proximal operator evaluation. We also detail how to handle a quadratic regularizer within each term's prox operator, which has a closed form in terms of the unregularized prox operator.

**Initialization** Instead of setting $g_i^0 = f_i'(x^0)$ before commencing the algorithm, we recommend using $g_i^0 = 0$ instead. This avoids the cost of a initial pass over the data. In practical effect this is similar to the SDCA initialization of each dual variable to 0.

# 5 Experiments

We tested our algorithm which we call Point-SAGA against SAGA [Defazio et al., 2014a], SDCA [Shalev-Shwartz and Zhang, 2013a], Pegasos/SGD [Shalev-Shwartz et al., 2011] and the catalyst acceleration scheme [Lin et al., 2015]. SDCA was chosen as the inner algorithm for the catalyst scheme as it doesn't require a step-size, making it the most practical of the variants. Catalyst applied to SDCA is essentially the same algorithm as proposed in Shalev-Shwartz and Zhang [2013c]. A single inner epoch was used for each SDCA invocation. Accelerated MISO as well as the primal-dual FIG method [Lan and Zhou, 2015] were excluded as we wanted to test on sparse problems and they are not designed to take advantage of sparsity. The step-size parameter for each method ($\kappa$ for catalyst-SDCA) was chosen using a grid search of powers of 2. The step size that gives the lowest error at the final epoch is used for each method.

We selected a set of commonly used datasets from the LIBSVM repository [Chang and Lin, 2011]. The pre-scaled versions were used when available. Logistic regression with $L_2$ regularization was applied to each problem. The $L_2$ regularization constant for each problem was set by hand to ensure $f$ was not in the big data regime $n \geq L/\mu$; as noted above, all the methods perform essentially the same when $n \geq L/\mu$. The constant used is noted beneath each plot. Open source code to exactly replicate the experimental results is available at https://github.com/adefazio/point-saga.

**Algorithm scaling with respect to** $n$ The key property that distinguishes accelerated FIG methods from their non-accelerated counterparts is their performance scaling with respect to the dataset size. For large datasets on well-conditioned problems we expect from the theory to see little difference between the methods. To this end, we ran experiments including versions of the datasets subsampled randomly without replacement in 10% and 5% increments, in order to show the scaling with $n$ empirically. The same amount of regularization was used for each subset.

Figure 1 shows the function value sub-optimality for each dataset-subset combination. We see that in general accelerated methods dominate the performance of their non-accelerated counter-parts. Both SDCA and SAGA are much slower on some datasets comparatively than others. For example, SDCA is very slow on the 5 and 10% COVTYPE datasets, whereas both SAGA and SDCA are much slower than the accelerated methods on the AUSTRALIAN dataset. These differences reflect known properties of the two methods. SAGA is able to adapt to inherent strong convexity while SDCA can be faster on very well-conditioned problems.

There is no clear winner between the two accelerated methods, each gives excellent results on each problem. The Pegasos (stochastic gradient descent) algorithm with its slower than linear rate is a clear loser on each problem, almost appearing as an almost horizontal line on the log scale of these plots.

**Non-smooth problems** We also tested the RCV1 dataset on the hinge loss. In general we did not expect an accelerated rate for this problem, and indeed we observe that Point-SAGA is roughly as fast as SDCA across the different dataset sizes.

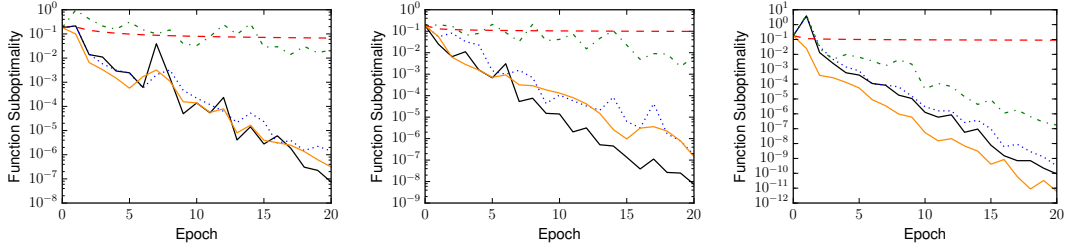

(a) COVTYPE $\mu = 2 \times 10^{-6}$ : 5%, 10%, 100% subsets

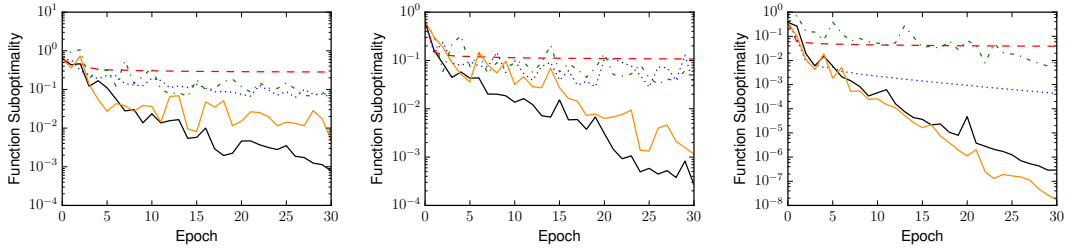

(b) AUSTRALIAN $\mu = 10^{-4}$: 5%, 10%, 100% subsets

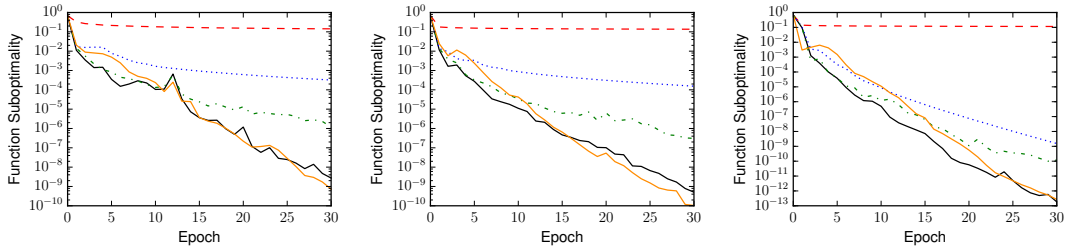

(c) MUSHROOMS $\mu = 10^{-4}$: 5%, 10%, 100% subsets

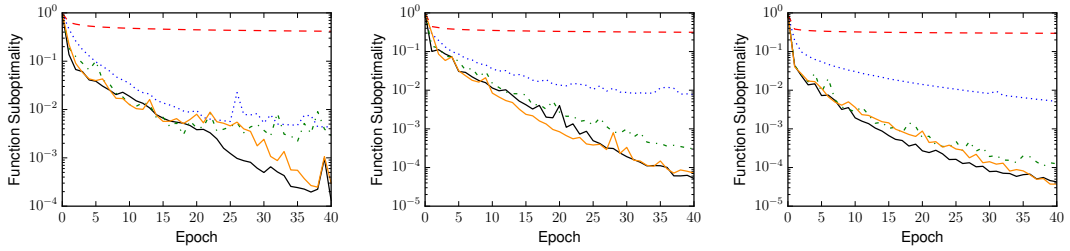

(d) RCV1 with hinge loss, $\mu = 5 \times 10^{-5}$: 5%, 10%, 100% subsets

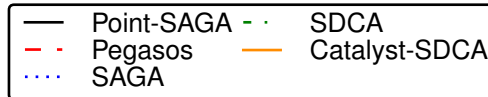

Figure 1: Experimental results

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
