[Supplementary Material]

# Appendix: A Simple Practical Accelerated Method for Finite Sums

**Aaron Defazio**
Ambiata, Sydney Australia

## 1   Proximal operators

For the most common binary classification and regression methods, implementing the proximal operator is straight-forward. In this section let $y_j$ be the label or target for regression, and $X_j$ the data instance vector. We assume for binary classification that $y_j \in \{-1, 1\}$.

**Hinge loss:**
$$f_j(z) = l(z; y_j, X_j) = \max\{0, \ 1 - y_j \langle z, X_j \rangle\}.$$
The proximal operator has a closed form expression:
$$\text{prox}_{\gamma f_j}(z) = z - \gamma y_j \nu X_j,$$
where:
$$s = \frac{1 - y_j \langle z, X_j \rangle}{\gamma \|X_j\|^2}.$$
$$\nu = \begin{cases} -1 & s \geq 1 \\ 0 & s \leq 0 \\ -s & \text{otherwise} \end{cases}.$$

**Logistic loss:**
$$f_j(z) = l(z; y_j, X_j) = \log\left(1 + \exp\left(-y_j X_j^T z\right)\right).$$
There is no closed form expression, however it can be computed very efficiently using Newton iteration, since it can be reduced to a 1D minimization problem. In particular, let $c_0 = 0$, $\gamma' = \gamma \|X_j\|^2$, and $a = \langle z, X_j \rangle$. Then iterate until convergence:
$$s^k = \frac{-y_j}{1 + \exp\left(y_j c^k\right)},$$
$$c^{k+1} = c^k - \frac{\gamma' s^k + c^k - a}{1 - y' s^k - \gamma' s^k s^k}.$$
The prox operator is then $\text{prox}_{\gamma f_j}(z) = z - \left(a - c^k\right) X_j / \|X_j\|^2$. Three iterations are generally enough, but ill-conditioned problems or large step sizes may require up to 12. Correct initialization is important, as it will diverge when initialized with a point on the opposite side of 0 from the solution.

**Squared loss:**
$$f_j(z) = l(z; y_j, X_j) = \frac{1}{2}\left(X_j^T z - y_j\right)^2.$$
Let $\gamma' = \gamma \|X_j\|^2$ and $a = \langle z, X_j \rangle$. Define:
$$c = \frac{a + \gamma' y}{1 + \gamma'}.$$
Then $\text{prox}_{\gamma f_j}(z) = z - (a - c) X_j / \|X_j\|^2$.

**L2 regularization**

Including a regularizer within each $f_i$, i.e. $F_i(x) = f_i(x) + \frac{\mu}{2} \|x\|^2$, can be done using the proximal operator of $f_i$. Define the scaling factor:

$$\rho = 1 - \frac{\mu\gamma}{1 + \mu\gamma}.$$

Then $\text{prox}_{\gamma F_i}(z) = \text{prox}_{\rho\gamma f_i}(\rho z)$.

## 2 Proofs

**Lemma 1.** *Under Algorithm 1, taking the expectation over the random choice of $j$, conditioning on $x^k$ and each $g_i^k$, allows us to bound the following inner product at step $k$:*

$$E\left\langle \gamma\left[g_j^k - \frac{1}{n}\sum_{i=1}^n g_i^k\right] - \gamma g_j^*, \left(x^k - x^*\right) + \gamma\left[g_j^k - \frac{1}{n}\sum_{i=1}^n g_i^k\right] - \gamma g_j^* \right\rangle$$

$$\leq \gamma^2 \frac{1}{n}\sum_{i=1}^n \left\|g_i^k - g_i^*\right\|^2.$$

*Proof.* We start by splitting on the right hand side of the inner product:

$$= E\left\langle \gamma\left[g_j^k - \frac{1}{n}\sum_{i=1}^n g_i^k\right] - \gamma g_j^*, \, x^k - x^* \right\rangle$$

$$+ E\left\langle \gamma\left[g_j^k - \frac{1}{n}\sum_{i=1}^n g_i^k\right] - \gamma g_j^*, \, \gamma\left[g_j^k - \frac{1}{n}\sum_{i=1}^n g_i^k\right] - \gamma g_j^* \right\rangle \quad (1)$$

The first inner product has expectation 0 on the left hand side (Recall that $E[g_j^*] = 0$), so it's simply 0 in expectation (we may take expectation on the left since the right doesn't depend on $j$). The second inner product is the same on both sides, so we may convert it to a norm-squared term. So we have:

$$= \gamma^2 E\left\| g_j^k - \frac{1}{n}\sum_{i=1}^n g_i^k - g_j^* \right\|^2$$

$$\leq \gamma^2 E\left\|g_j^k - g_j^*\right\|^2 = \gamma^2 \frac{1}{n}\sum_{i=1}^n \left\|g_i^k - g_i^*\right\|^2.$$

The inequality used is just an application of the variance formula $E[(X - E[X])^2] = E[X^2] - E[X]^2 \leq E[X^2]$. $\qquad\square$

**Corollary 2.** *Chaining the main theorem gives a convergence rate for point-saga at step $k$ under the constants given in of:*

$$E\left\|x^k - x^*\right\|^2 \leq (1 - \kappa)^k \frac{\mu + L}{\mu} \left\|x^0 - x^*\right\|^2,$$

*if each $f_i : \mathbb{R}^d \to \mathbb{R}$ is L-smooth and $\mu$-strongly convex.*

*Proof.* First we simplify $T^0$ using $c = 1/\mu L$ and use Lipschitz smoothness:

$$T^0 = \frac{1}{\mu L} \cdot \frac{1}{n}\sum_i \left\|g_i^0 - g_i^*\right\|^2 + \left\|x^0 - x^*\right\|^2$$

$$\leq \frac{L}{\mu} \cdot \left\|x^0 - x^*\right\|^2 + \left\|x^0 - x^*\right\|^2$$

$$= \frac{\mu + L}{\mu} \left\|x^0 - x^*\right\|^2.$$

Now recall that the main theorem gives a bound $E\left[T^{k+1}\right] \leq (1-\kappa)\,T^k$ where the expectation is conditional on $x^k$ and each $g_i^k$ from step $k$, taking expectation over the randomness in the choice of $j$. We can further take expectation with respect to $x^k$ and each $g_i^k$, giving the unconditional bound:

$$E\left[T^{k+1}\right] \leq (1-\kappa)\,E\left[T^k\right].$$

Chaining over $k$ gives the result. $\qquad\qquad\qquad\qquad\qquad\qquad\qquad\qquad\qquad\square$

**Theorem 3.** *Suppose each $f_i : \mathbb{R}^d \to \mathbb{R}$ is $\mu$-strongly convex, $\left\|g_i^0 - g_i^*\right\| \leq B$ and $\left\|x^0 - x^*\right\| \leq R$. Then after $k$ iterations of Point-SAGA with step size $\gamma = R/B\sqrt{n}$:*

$$E\left\|\bar{x}^k - x^*\right\|^2 \leq 2\,\frac{\sqrt{n}\,(1 + \mu\,(R/B\sqrt{n}))}{\mu k}\,RB,$$

*where $\bar{x}^k = \frac{1}{k} E \sum_{t=1}^k x^t$.*

*Proof.* Recall the bound on the Lyapunov function established in the main theorem:

$$E\left[T^{k+1}\right] \leq T^k + \left(\alpha\gamma^2 - \frac{c}{n}\right) \frac{1}{n} \sum_i^n \left\|g_i^k - g_i^*\right\|^2$$

$$+ \left(\frac{c}{n} - \alpha\gamma^2 - \frac{\alpha\gamma}{L}\right) E\left\|g_j^{k+1} - g_j^*\right\|^2$$

$$- \kappa E\left\|x^k - x^*\right\|^2.$$

In the non-smooth case this holds with $L = \infty$. In particular, if we take $c = \alpha\gamma^2 n$, then:

$$-\kappa E\left\|x^{k+1} - x^*\right\|^2 \geq E\left[T^{k+1}\right] - T^k.$$

Recall that this expectation is (implicitly) conditional on $x^k$ and each $g_i^k$ from step $k$, Taking expectation over the randomness in the choice of $j$. We can further take expectation with respect to $x^k$ and each $g_i^k$, and negate the inequality, giving the unconditional bound:

$$\kappa E\left\|x^{k+1} - x^*\right\|^2 \leq E\left[T^k\right] - E\left[T^{k+1}\right].$$

We now sum this over $t = 0 \ldots k$:

$$\kappa E \sum_{t=1}^k \left\|x^t - x^*\right\|^2 \leq T^0 - E\left[T^k\right].$$

We can drop the $-E\left[T^k\right]$ since it is always negative. Dividing through by $k$:

$$\frac{1}{k} E \sum_{t=1}^k \left\|x^t - x^*\right\|^2 \leq \frac{1}{\kappa k}T^0.$$

Now using Jensen's inequality on the left gives:

$$E\left\|\bar{x}^k - x^*\right\|^2 \leq \frac{1}{\kappa k}T^0,$$

where $\bar{x}^k = \frac{1}{k} E \sum_{t=1}^k x^t$. Now we plug in $T^0 = \frac{c}{n} \sum_i \left\|g_i^0 - g_i^*\right\|^2 + \left\|x^0 - x^*\right\|^2$ with $c = \alpha\gamma^2 n \leq \gamma^2 n$:

$$E\left\|\bar{x}^k - x^*\right\|^2 \leq \frac{\gamma^2 n}{\kappa k}\frac{1}{n} \sum_i \left\|g_i^0 - g_i^*\right\|^2 + \frac{1}{\kappa k}\left\|x^0 - x^*\right\|^2.$$

Now we plug in the bounds in terms of $B$ and $R$:

$$E\left\|\bar{x}^k - x^*\right\|^2 \leq \frac{\gamma^2 n}{\kappa k}B^2 + \frac{1}{\kappa k}R^2.$$

In order to balance the terms on the right, we need:

$$\frac{\gamma^2 n}{\kappa k}B^2 = \frac{1}{\kappa k}R^2,$$

$$\therefore \gamma^2 n B^2 = R^2,$$

$$\therefore \gamma^2 = \frac{R^2}{nB^2}.$$

So we can take $\gamma = R/B\sqrt{n}$, giving a rate of:

$$
\begin{aligned}
E\left\|\bar{x}^k - x^*\right\|^2 &\leq \frac{2}{\kappa k}R^2 \\
&= 2\frac{1+\mu\gamma}{\mu\gamma k}R^2 \\
&= 2\frac{\sqrt{n}\left(1+\mu\left(R/B\sqrt{n}\right)\right)}{\mu k}RB.
\end{aligned}
$$

$\square$

# 3 Proximal operator bounds

In this section we rehash some simple bounds from proximal operator theory that we will use in this work. Define the short-hand $p_{\gamma f}(x) = \mathrm{prox}_{\gamma f}(x)$, and let $g_{\gamma f}(x) = \frac{1}{\gamma}\left(x - p_{\gamma f}(x)\right)$, so that $p_{\gamma f}(x) = x - \gamma g_{\gamma f}(x)$. Note that $g_{\gamma f}(x)$ is a subgradient of $f$ at the point $p_{\gamma f}(x)$. This relation is known as the optimality condition of the proximal operator.

We will also use a few standard convexity bounds without proof. Let $f : \mathbb{R}^d \to \mathbb{R}$ be a convex function with strong convexity constant $\mu \geq 0$ and Lipschitz smoothness constant $L$. Let $x^*$ be the minimizer of $f$, then for any $x, y \in \mathbb{R}^d$:

$$\langle f'(x) - f'(y), x - y \rangle \geq \mu \left\|x - y\right\|^2, \tag{2}$$

$$\left\|f'(x) - f'(y)\right\|^2 \leq L^2 \left\|x - y\right\|^2. \tag{3}$$

**Proposition 4.** *(Firm non-expansiveness) For any $x, y \in \mathbb{R}^d$, and any convex function $f : \mathbb{R}^d \to \mathbb{R}$ with strong convexity constant $\mu \geq 0$,*

$$\langle x - y, p_{\gamma f}(x) - p_{\gamma f}(y) \rangle \geq (1 + \mu\gamma)\left\|p_{\gamma f}(x) - p_{\gamma f}(y)\right\|^2.$$

*Proof.* Using strong convexity of $f$, we apply Equation 2 at the (sub-)gradients $g_{\gamma f}(x)$ and $g_{\gamma f}(y)$, and their corresponding points $p_{\gamma f}(x)$ and $p_{\gamma f}(y)$:

$$\langle g_{\gamma f}(x) - g_{\gamma f}(y), p_{\gamma f}(x) - p_{\gamma f}(y) \rangle \geq \mu \left\|p_{\gamma f}(x) - p_{\gamma f}(y)\right\|^2.$$

We now multiply both sides by $\gamma$, then add $\left\|p_{\gamma f}(x) - p_{\gamma f}(y)\right\|^2$ to both sides:

$$\langle p_{\gamma f}(x) + \gamma g_{\gamma f}(x) - p_{\gamma f}(y) - \gamma g_{\gamma f}(y), p_{\gamma f}(x) - p_{\gamma f}(y) \rangle \geq (1 + \mu\gamma)\left\|p_{\gamma f}(x) - \mathrm{p}_{\gamma f}(y)\right\|^2,$$

leading to the bound by using the optimality condition: $p_{\gamma f}(x) + \gamma g_{\gamma f}(x) = x$. $\square$

**Proposition 5.** *(Moreau decomposition) For any $x \in \mathbb{R}^d$, and any convex function $f : \mathbb{R}^d \to \mathbb{R}$ with Fenchel conjugate $f^*$ :*

$$p_{\gamma f}(x) = x - \gamma p_{\frac{1}{\gamma}f^*}(x/\gamma). \tag{4}$$

*Recall our definition of $g_{\gamma f}(x) = \frac{1}{\gamma}\left(x - p_{\gamma f}(x)\right)$ also. After combining, the following relation thus holds between the proximal operator of the conjugate $f^*$ and $g_{\gamma f}$:*

$$p_{\frac{1}{\gamma}f^*}(x/\gamma) = \frac{1}{\gamma}\left(x - p_{\gamma f}(x)\right) = g_{\gamma f}(x). \tag{5}$$

*Proof.* Let $u = p_{\gamma f}(x)$, and $v = \frac{1}{\gamma}(x - u)$. Then $v \in \partial f(u)$ by the optimality condition of the proximal operator of $f$ (namely if $u = p_{\gamma f}(x)$ then $u = x - \gamma v \Leftrightarrow v \in \partial f(u)$). It follows by conjugacy of $f$ that $u \in \partial f^*(v)$. Thus we may interpret $v = \frac{1}{\gamma}(x - u)$ as the optimality condition of a proximal operator of $f^*$ :

$$v = p_{\frac{1}{\gamma} f^*}\left(\frac{1}{\gamma} x\right).$$

Plugging in the definition of $v$ then gives:

$$\frac{1}{\gamma}(x - u) = p_{\frac{1}{\gamma} f^*}\left(\frac{1}{\gamma} x\right).$$

Further plugging in $u = p_{\gamma f}(x)$ and rearranging gives the result. $\qquad\square$