[Reviews · NeurIPS 2016]

Reviewer 1

Summary

This paper presents an accelerated variant of SAGA for finite sums. Accelerated algorithms are part of an important class of optimization methods which have a much better dependency to the condition number. There is recently been a lot of renewed interest in this class of methods and many papers have been published including accelerated SDCA or the Catalyst method. The authors here describe an accelerated version for SAGA in the regime when L/mu > n. They describe their method as “significantly simpler and requires less tuning than existing accelerated methods”.

Qualitative Assessment

Technical quality: To the best of my knowledge, the paper is technically sound and the proofs derived in the paper are correct and not hard to follow. There is not much intuition given as to why the modified step leads to an acceleration. I know accelerated methods are not always very intuitive but is there any intuition that could be gained from recent work on accelerated methods? e.g. the work of Allen-Zhu, Zeyuan, and Lorenzo Orecchia. "Linear coupling: An ultimate unification of gradient and mirror descent.", where the acceleration results from a primal-dual point of view on the progress of the algorithm. Novelty/originality: The main problem with the paper is the novelty aspect of the work. Similar convergence rates for accelerated variance reduced SGD have already been derived in the catalyst paper as well as other references. One missing reference is Nitanda, Atsushi. "Accelerated Stochastic Gradient Descent for Minimizing Finite Sums." arXiv preprint arXiv:1506.03016 (2015). The authors claim that their method is “significantly simpler and requires less tuning than existing accelerated methods” but I think this point is quite controversial as one could claim that other methods also only have one parameter to tune. The authors acknowledge in the experimental section that “SDCA was chosen as the inner algorithm for the catalyst scheme as it doesn’t require a step-size, making it the most practical of the variants”. Furthermore, the only parameter described for the other baselines in the experimental section is the step-size (“The step-size parameter for each method (κ for catalyst-SDCA) was chosen using a grid search of powers of 2.“). I think this point is very ambiguous in the paper and needs to be rewritten with some compelling arguments as to why their method is simpler or this claim should be toned down. One positive contribution though is that the method proposed by the authors handles non-smooth loss which is not the case for most existing variance-reduced methods (expect SDCA). Could the authors elaborate on whether or not Theorem 7 guarantees a better convergence in comparison to SDCA? Experimental section: In the experiments with the hinge loss (as a non-smooth objective), the convergence seems linear while their Theorem 7 provides a sub-linear convergence. Could the authors comment on a potential improvement for “nearly everywhere smooth loss functions” as done in the SDCA paper? Minor: Lines 68-9: there is a typo where the sign of g_j^k is negative.

Confidence in this Review

3-Expert (read the paper in detail, know the area, quite certain of my opinion)


Reviewer 2

Summary

This paper proposed an accelerated and incremental proximal point method for solving finite-sums problems. Theoretical analysis showed the convergence rate of this method has a square-root dependence on the condition number. And its numerical performance is superior to non-accelerated methods on ill-conditioned problems.

Qualitative Assessment

It is good to see that this paper extends SAGA to the proximal point version and accelerates this method. However, the motivation of studying this kind of extension is not well justified. The algorithm proposed in this paper requires that each component function has an easy proximal operator. However, this actually rules out a lot of important applications. The logistic regression does not satisfy this requirement. As the authors showed in the supplementary material, the logistic loss does not have easy proximal mapping, whose solution needs to be computed by a Newton method, which could be costly. The experimental part is poorly written. First, I cannot even find what problem is testing in this part. Second, “epoch” was used as the x-axis in the figures, but how is an epoch defined? Since many different methods are involved, it would be important to clarify it. The claim in line 48-49 is misleading. Your method does not fall into the lower bound of [Lan and Zhou 2015] because the proximal operator is used here, which implicitly utilizes the gradient information of the next iterate. However, in their analysis only the gradient of the current iterate is accessible. The equation between line 68-69 is wrong. There should be a minus sign before g^k_j according to your algorithm description.

Confidence in this Review

3-Expert (read the paper in detail, know the area, quite certain of my opinion)


Reviewer 3

Summary

The authors propose a novel accelerated optimization scheme for finite-sum objective functions as they appear in (regularized) empirical risk minimization. The approach is closely related to existing methods, most prominently SAGA. An accelerated rate of convergence is proven. In contrast to its competitors the method is even applicable to non-smooth objectives (e.g., SVM training with the hinge loss). The only downside is the application of the prox operator, which can be a limiting factor in some cases, but actually not so for standard supervised learning. The emphasis of the paper is on establishing the theoretical analysis, which is followed by a brief but meaningful experimental analysis.

Qualitative Assessment

The paper reads very well, even the proofs. Since I don't have a lot of experience with this type of analysis I did not follow every detail, but the authors guide the reader very nicely through the proofs. This does not mean that the paper is trivial to follow, but provided the depth of the material the authors do a great job. The experimental evaluation is focused on validating the predictions of the theory. For the empirical assessment the use of machine training problems as benchmarks is not essential, other problems would have done the job. The rates are relevant to optimization in the first place, but for machine learning they are only one aspect (e.g., dependency of the training time on hyperparameter settings can be a game changer). Such aspects have been ignored. This becomes apparent, e.g., from the fact that the regularization constant was chosen so that differences between the methods show up nicely, and not to minimize the prediction (e.g., cross validation) error. The experimental evaluation is still convincing in the sense that the method seems to suitable for fast training of learning machines.

Confidence in this Review

2-Confident (read it all; understood it all reasonably well)


Reviewer 4

Summary

The authors propose and analyze an accelerated incremental optimization method for finite sum. The algorithm is similar to SAGA with the difference that proximal operations are used in lieu of gradients steps.

Qualitative Assessment

The authors propose an algorithm, Point-SAGA, that is an accelerated incremental minimization method based on proximal operations (i.e. implicit gradients) instead of (explicit) gradients. Convergence rates are derived in the case where the functions are strongly convex (dual property to smoothness) and linear rates are given when they are in addition smooth. The algorithm is original and the analysis globally convincing however: 1) The writing has to be improved notably for the experiments/implementation; as well as for the "operators" and contractions definitions. 2) The reach of the algorithm seems to be limited as functions have to be prox simple and strongly convex. Although prox can be solved by inner loops (similarly to Catalyst), the complexity can then be a burden; and popular regularizations as l1 norms or projections are out of reach of the algorithm. Particular Remarks: * The authors mention l 38-40 that "the sum of gradients can be cached and updated efficiently at each step, and in most cases instead of storing a full vector, only a single real value needs to be stored." This is unclear for me as step 3 of the authors algorithm necessitates replacing only the j-th gradient which means that the j-th gradient at time k has to be accesible and not only the sum of the gradients. * A reference should be given for Prop 1. * The term Firm Non-Expansiveness is used in a confusing manner. In Prop. 1, named Firm Non-Expansiveness, the verified inequality means that p_{gamma f} is (1+mu*gamma)-cocoercive or that (1+mu*gamma)*p_{gamma f} is FNE but not p_{gamma f} directly. * Lemma 4, the conditioning with respect to the past should be clarified e.g. by writing it E[ X | \mathcal{F}_k]; same thing for Th. 5. In addition, the inequality used for proving Lemma 4 (1 in the appendix) can appear strange at first glance but is simply the Variance/Expectation usual formula; the proof could be clarified by stating it. * Experiments. The authors seem to deal with L2 regularized logistic regression although it is not directly stated! The comparison with catalyst is then the most significant as both Catalyst and Point-Saga have double loops for log. reg., then why not taking Newton iterations as an inner scheme for both? Also, if possible it would be nice for comparison to also have comparison with full batch methods on subsets (e.g. PPA and ADMM). Minor remarks: * l. 112 \sum_j^n should be \sum_{j=1}^n * There is a equation numbering issue for Eqs. (4)-(5). They correspond to the same equality, and the "first inner product in Eq. (5)" actually Eq. (4)... * The term "non-sparse updates" l 169 is confusing; lease reformulate

Confidence in this Review

3-Expert (read the paper in detail, know the area, quite certain of my opinion)


Reviewer 5

Summary

This paper proposes a SAGA-like extension of proximal point algorithm for minimizing finite sums. The paper shows the optimal convergence complexity for smooth strongly convex problems with a relatively simple proof. The method has only a single hyper-parameter (i.e. learning rate), so that it is more practical than other optimal methods. Experiments show competitive performance compared to some optimal methods.

Qualitative Assessment

The paper is well written and technical part may be correct. Although the paper is incremental, it makes the following contributions: - A highly practical method; it has only a single hyper-parameter. # Added: However, the method has limited applicability, e.g., needs Newton method for logistic regression. - The method achieves optimal complexity with a simple proof, although a proof of an optimal method tend to be complex in this literature. Thus, the paper is useful to the literature.

Confidence in this Review

2-Confident (read it all; understood it all reasonably well)


Reviewer 6

Summary

This paper presents an accelerated version of the SAGA algorithm, Point-SAGA, for minimizing finite sums. This randomized algorithm is an extension of the SAGA algorithm and achieves an accelerated convergence rate comparable to that of e.g. Catalyst-accelerated SDCA. It is based on gradient and prox oracle access and can be applied to non-smooth problems. After motivating the desirability of accelerated algorithms and comparing their method with SAGA, the authors prove the convergence rate of Point-SAGA for strongly convex functions both in the smooth and non-smooth case. Next, they very briefly touch upon a few implementation details and conclude with an empirical comparison of their algorithm against several other commonly used optimization algorithms.

Qualitative Assessment

This paper addresses its stated goal of providing an accelerated algorithm for finite sum optimization. The algorithm is somewhat simpler to state and analyze than other accelerated methods for finite sums that I am aware of, but it does not seem to yield substantively better guarantees or empirical results than existing algorithms (its guarantees and empirical performance are roughly matched by, e.g. Catalyst-accelerated SDCA). It represents a step forward in terms of ease of implementation (and it is indeed quite simple) but that is it as far as I am aware. The paper is clearly communicated in general, although there are one or two steps in proofs which are slightly opaque (e.g. the proof of corollary 6 makes use of the smoothness property of the functions f_i, which is never explicitly defined in the paper, and could be confusing for some). I believe there is a small bug in the proof of Theorem 5, in particular, the setting of c and \gamma on line 141, which do not seem to guarantee that the second coefficient in the equation on the line above is non-positive. At the very least, there must be conditions on L and \mu for the statement to be true (as a counterexample, take a spherical quadratic where L=\mu, meaning \gamma=0 and the second term is strictly positive, furthermore \alpha = 1 so there is no convergence). This is probably not a fatal flaw, in that I believe there is probably a choice of \gamma such that both coefficients are negative and \alpha is strictly smaller than 1, but it is somewhat concerning since the convergence guarantee rests on it.

Confidence in this Review

2-Confident (read it all; understood it all reasonably well)